Predicting maintenance lithium response for bipolar disorder from electronic health records—a retrospective study

http://orcid.org/0000-0003-2286-3862 Hayes Joseph F. 1 2
http://orcid.org/0000-0001-7866-143X Ben Abdesslem Fehmi 1 3 4
Eloranta Sandra 5
Osborn David P. J. 1 2
http://orcid.org/0000-0001-7949-1815 Boman Magnus 1 5 6 m.boman@ucl.ac.uk
1 Department of Psychiatry, University College London, University of London , London , United Kingdom
2 Camden and Islington NHS foundation Trust , London , United Kingdom
3 Research Institutes of Sweden (RISE) , Stockholm , Sweden
4 Department of Clinical Neuroscience, Karolinska Institutet , Stockholm , Sweden
5 Division of Clinical Epidemiology, Department of Medicine Solna, Karolinska Institutet , Stockholm , Sweden
6 MedTechLabs, BioClinicum, Karolinska University Hospital , Stockholm , Sweden
Chen Chong
Electronic publication date: 2024 Oct 14
Publication date: 2024
Volume: 12
Electronic Location ID: e17841
Received 2023 Oct 30; Accepted 2024 Jul 10
Copyright: © 2024 Hayes et al.
Copyright year: 2024
Copyright holder: Hayes et al.
License: This is an open access article distributed under the terms of the Creative Commons Attribution License, which permits unrestricted use, distribution, reproduction and adaptation in any medium and for any purpose provided that it is properly attributed. For attribution, the original author(s), title, publication source (PeerJ) and either DOI or URL of the article must be cited.
License URL: https://creativecommons.org/licenses/by/4.0/

Keywords: Bipolar disorder, Lithium, Maintenance response prediction, Machine learning, Retrospective study

Funding: Wellcome Trust 211085/Z/18/Z UK Research and Innovation MR/V023373/1 University College London Hospitals NIHR Biomedical Research Centre NIHR North Thames Applied Research Collaboration UK Research and Innovation Medical Research Council MR/W014386/1 This study was supported by grant 211085/Z/18/Z from the Wellcome Trust. Joseph F. Hayes and David PJ Osborn are supported by the UK Research and Innovation grant MR/V023373/1, the University College London Hospitals NIHR Biomedical Research Centre, and the NIHR North Thames Applied Research Collaboration. David PJ Osborn is also funded by UK Research and Innovation Medical Research Council Grant MR/W014386/1. The funders had no role in study design, data collection and analysis, decision to publish, or preparation of the manuscript.

==============================
Background

Optimising maintenance drug treatment selection for people with bipolar disorder is challenging. There is some evidence that clinical and demographic features may predict response to lithium. However, attempts to personalise treatment choice have been limited.

Method

We aimed to determine if machine learning methods applied to electronic health records could predict differential response to lithium or olanzapine. From electronic United Kingdom primary care records, we extracted a cohort of individuals prescribed either lithium (19,106 individuals) or olanzapine (12,412) monotherapy. Machine learning models were used to predict successful monotherapy maintenance treatment, using 113 clinical and demographic variables, 8,017 (41.96%) lithium responders and 3,831 (30.87%) olanzapine responders.

Results

We found a quantitative structural difference in that lithium maintenance responders were weakly predictable in our holdout sample, consisting of the 5% of patients with the most recent exposure. Age at first diagnosis, age at first treatment and the time between these were the most important variables in all models.

Discussion

Even if we failed to predict successful monotherapy olanzapine treatment, and so to definitively separate lithium vs. olanzapine responders, the characterization of the two groups may be used for classification by proxy. This can, in turn, be useful for establishing maintenance therapy. The further exploration of machine learning methods on EHR data for drug treatment selection could in the future play a role for clinical decision support. Signals in the data encourage further experiments with larger datasets to definitively separate lithium vs. olanzapine responders.

Introduction

First-line medications for the maintenance treatment of bipolar disorder (BPD) include lithium, anticonvulsant and antipsychotic medications, with lithium having evidence for being the most effective (Yatham et al., 2018). Of those treated with lithium, 30% have an enduring full response and at least 30% of patients are non-responsive (Garnham et al., 2007; Alda, 2017). Those patients who stabilise on lithium especially well can cease further mood episodes and return to pre-illness levels of function (Tondo et al., 2019). It is unclear what proportion of patients respond well to other maintenance treatment options. Response to antipsychotics has been found to be heterogeneous in studies of psychosis (Perkins et al., 2004), but a full responder phenotype has not been identified for other maintenance treatments for BPD. Optimising treatment selection for an individual could significantly improve outcomes including relapse, morbidity and mortality. During the period covered by this study, the National Institute for Health and Care Excellence (UK) recommended lithium, sodium valproate or olanzapine as first-line maintenance monotherapy for BPD (National Collaborating Centre for Mental Health, 2014). In-line with this guidance we have found that olanzapine was the most commonly prescribed antipsychotic to people with bipolar disorder (Ng et al., 2021). More recent and international treatment guidelines, for example the Canadian Network for Mood and Anxiety Treatments and International Society for Bipolar Disorders (Yatham et al., 2018), and the British Association for Psychopharmacology (Goodwin et al., 2016) also recommend olanzapine, but have included other antipsychotic medications as suitable maintenance mood stabilisers, but the number prescribed these was smaller in our cohort. We are aware of one randomised controlled trial that at 1 year found reduced risk of discontinuation (but no difference in relapse) when comparing olanzapine with lithium (Tohen et al., 2005). We did not consider sodium valproate as a comparison group because its use is now limited in women (and men) of childbearing age.

It is a widely held assumption that the lithium responder phenotype is classical in terms of its resemblance to the original Kraepelinian descriptions of manic depression (Alda, 2004). In a meta-analysis, (Hui et al., 2019) lithium response was associated with mania–depression polarity sequence, absence of psychotic symptoms, shorter duration of illness prior to lithium initiation, family history of BPD, and later onset of illness. Other features included low rates of psychiatric comorbidity, clinical course characterised by distinct episodes marked with good inter-episode functioning, and family history of lithium response. However, these features have been identified via research using univariable analysis, in small samples, often lacking comparison groups. This means that these features may predict a benign illness course, irrespective of choice of maintenance treatment. A more recent systematic review concluded that, when only high quality studies were included, no features predictive of good outcome during lithium treatment could be identified (Grillault Laroche et al., 2020). Since this meta-analysis and systematic review were published, studies have reported on the prediction of BPD treatment response using clinical features and machine learning (ML) (Kim et al., 2019; Nunes et al., 2020a). One identified attention-deficit hyperactivity disorder comorbidity, non-suicidal self-injurious behaviour, severity of mania, comorbid anxiety disorder and suicide risk as predictive features of lithium response. This can be contrasted with a model to predict quetiapine response which included employment, agoraphobia, irritability, cannabis use and depression (Kim et al., 2019). This study was limited by its small sample size ( 240 lithium-treated patients and 242 treated with quetiapine) and short duration (6 months). A second study included a cohort of 1,266 lithium-treated patients from multiple sites around the world, with a minimum treatment duration of 1 year. This study found high heterogeneity in predictors from different sites but variables characterising the pre-treatment clinical course were among the most important. This study was limited by its lack of a comparison group, like most of the previous literature (Nunes et al., 2020a). A recent study used a combination of genetic and clinical data, letting polygenic risk scores stratify patients (n = 1,034) first and then predicting lithium response (Cearns et al., 2022). This research followed positive findings from GWAS studies regarding lithium response prediction for patients with BPD (Gordovez & McMahon, 2020), albeit concluding that more research was needed into validation of the models and the amount of variance in response that was explained by these models (Oraki Kohshour et al., 2021). We answer this call by investigating a large sample of patients in a quantitative model, employing a large number of supervised machine learning (ML) methods. In addition, we let unsupervised methods for clustering various subpopulations inform our supervised approach to build a model that generalises maximally to other populations (Carbonell, Boman & Laukka, 2021).

Many of the previously identified clinical predictors of lithium response are available at scale in electronic health records (EHRs). EHRs have grown toward comprehensive coverage in many countries. If it was possible to generate personalised predictions for treatment response from this data and share them with clinicians, it would result in direct improvements for patient care (Torous & Baker, 2016). Given the heterogeneity in lithium treatment response and the decline in its use in many countries (because of the close laboratory monitoring required, and the risk of adverse effects), guidance and tools which increase probability of good response would be a major asset for prescribers and patients (Volkmann, Bschor & Köhler, 2020; Öhlund et al., 2018). Our aim was to determine if ML models could differentially predict response to lithium monotherapy vs. olanzapine monotherapy in an incident cohort of people with BPD. We defined response as being maintained on the same monotherapy for 2 years or more.

Materials and Methods

Study setting

We used a cohort of people with BPD from UK primary care EHRs to predict lithium vs. olanzapine maintenance treatment response using an ML model. Such models have a potential for identifying undiscovered subpopulations of patients with specific physiology and prognoses (Stevens et al., 2020). We chose olanzapine maintenance treatment as our comparison group as olanzapine is included in several guidelines as an effective maintenance treatment for BPD. Since olanzapine is often prescribed for acute mood stabilisation it is therefore straightforward to continue into the maintenance phase of treatment. We completed a cohort study using primary care EHR data collected between January 1, 2000, and December 31, 2018, by the Clinical Practice Research Datalink (CPRD) system. The study was approved by the Independent Scientific Advisory Committee of CPRD and the data was extracted in April 2019. SAC of CPRD protocol no.18316. CPRD Health Research Authority UK: East Midlands Derby Research Ethics Committee 21/EM/0265. CPRD obtains annual research ethics approval from the UK Health Research Authority Research Ethics Committee (East Midlands—Derby Research Ethics Committee reference number 05/MRE04/87) to receive and supply patient data for public health research. Therefore, no additional ethics approval was required for this study. CPRD collects de-identified patient data from a network of primary care practices across the UK. Primary care data are linked to a range of other health-related data to provide a longitudinal, representative UK population health dataset. At the time of data extraction, CPRD data encompassed 50 million patients, including 16 million currently registered patients. CPRD has two primary care data sets, CPRD Gold and CPRD Aurum. These have been shown to be broadly representative of the UK population (Wolf et al., 2019; Herrett et al., 2015). Data were collected as previously described in Hayes et al. (2016), Launders et al. (2022b). Specifically, within the UK National Health Service, primary care physicians are responsible for issuing prescriptions for ongoing medication use, so this information is well defined in the cohort. They also provide most long-term care to people with BPD, although the diagnosis is made in secondary care by a psychiatrist. The validity of severe mental illness diagnoses (including BPD) in primary care records has been established (Nazareth et al., 1993). The incidence rate of BPD in UK primary care databases is similar to that of other European cohorts (Hardoon et al., 2013). The data that support the findings of this study are available from CPRD but restrictions apply to the availability of these data, which were used under license for the current study, and so are not publicly available. Electronic health records are, by definition, considered to be sensitive data in the UK by the Data Protection Act and cannot be shared via public deposition because of information governance restriction in place to protect patient confidentiality. Access to data is available only once approval has been obtained through the individual constituent entities controlling access to the data. The primary care data can be requested via application to the CPRD.

Study population

The study base was all BPD patients prescribed lithium or olanzapine for the first time. All individuals 16 years or older with a diagnosis of BPD were included if they received two or more consecutive prescriptions for lithium or olanzapine treatment lasting 28 days or longer. At least a year of follow-up data before the first prescription of lithium or olanzapine as well as no previous record of receiving these medications was required to insure these were incident prescriptions. Patients with record of multiple drug exposures (i.e., more than one mood stabiliser including lithium, valproate, lamotrigine, carbamazepine, and any antipsychotic medication) continuously after the first 6 months of treatment were excluded. However, patients remained in the cohort if they received additional prescriptions for alternative mood stabilisers, antipsychotic medications, antidepressants or benzodiazepines in the first 6 months of treatment, to account for potential ongoing treatment of an acute mood event (washout period).

Outcomes

As our primary outcome, we defined treatment response as continuous prescription of the study drug (lithium or olanzapine) for at least 2 years without stopping treatment, switching to the other study drug or an alternative treatment, or without co-prescription of an additional treatment (excluding the aforementioned washout period). These alternative or additional BPD treatments were defined as valproate, lamotrigine, carbamazepine, any antipsychotic medication or any antidepressant medication. As a secondary outcome, we defined an equivocal response as receiving lithium monotherapy for greater than 1 year, but less than 2 years. This outcome attempts to capture both effectiveness and tolerability of the study drug and approximates the Lithium Response Phenotype Scale (Alda scale) (Scott et al., 2020) where recording of clinical improvement is not well quantified in the EHR, but because the secondary outcome proved too hard to predict with any significant accuracy, we will not report on it here.

We defined successful monotherapy maintenance treatment as treatment for at least 2 years, without switching medication or the addition of new psychotropic medications. Our definition therefore suggests the patient found the medication effective and tolerable. We believe this is a close approximation of treatment response as defined by the Alda Scale (Grof et al., 2002) and used in previous studies predicting lithium response. We see a similar proportion of “responders” in our cohort to earlier studies (Kim et al., 2019; Nunes et al., 2020a). The Alda scale is frequently dichotomized, with scores larger than six representing treatment response; this score is challenging to achieve if the duration of treatment is less than 2 years and if additional psychotropic medication is required. Therefore, our patients staying on treatment for over 2 years without additional medication will be similar to those identified as responders by the “gold standard” clinician rated scale (Nunes et al., 2020b). Given the complexity of the biology of successful lithium maintenance treatment (Papiol, Schulze & Heilbronner, 2022), we believe this makes our prediction results commensurable to the previous literature.

Data curation and data pre-processing

Since not only data quantity but also data quality is important, we carefully curated the data to be included. No data quality impairments due to data age (Pazzagli et al., 2018) were identified in the variables included. To account for possible confounding by indication when comparing outcome prediction for patients selected, we developed a propensity score for allocation of each of the two groups based on a widely defined set of 113 variables identified as candidates for being included in the ML models. Logistic regression was used to predict the propensity score. The two cohorts were trimmed to the range in which the propensity score overlapped whereby patients who had outlying propensity to be prescribed the different medications were excluded.

Agnostic feature selection and clustering

To investigate important associations between features, we extracted a number of demographic and clinical characteristics from the EHRs and used an agnostic approach to examine univariable association with treatment response. The rationale being that such associations assist in finding profiles for individuals doing very well (staying on the exposure drug for more than 2 years) and poorly (less than 1 year), as well as to eliminate redundant features prior to the modelling step. We employed a threshold of 0.05% of explained variance for first or secondary outcome, thereby pruning the long tail from 113 to 34 features. For binary variables we estimated the Spearman and Pearson correlation coefficients associated with the primary outcome (Fig. S1, Table S2) while for categorical variables, we performed a Chi-square test for equality in distribution in relation to treatment response (Fig. S2).

We subsequently investigated the conditional entropy for the variables we initially considered as machine learning features (Table S3), among these, 13 variables documented in the research literature (cf. Table 1), including the person’s age at exposure start, age at first diagnosis, number of years from first symptoms to first exposure, presence of psychosis, first presentation with depression or mania, the dominant mood polarity, sex, family history of BPD, weight group, and history of self-harm (Hui et al., 2019). The two first variables were highly correlated (Figs. S3 and S4). The 21 additional features that had predictive power but less support in the literature were also deemed relevant for further analyses (Fig. S5) and so assisted in indicating the relative feature importance between lithium and olanzapine responders. For this characterisation, we first clustered the lithium responders using unsupervised ML, reinvestigating the features employed for best results. In a second step, we inspected the stable clusters to form a description of what the automated procedure had found. We also carried out unsupervised ML analyses using UMAP for dimension reduction to explore how results clustered (McInnes, Healy & Melville, 2018). This approach is useful for establishing the K value of the K-means clustering, in our case K=2. Such clustering can elucidate the similarities between patients that respond very well to lithium, even if a lack of reliable markers to identify patients who might benefit the most from treatment have been reported.

Table 1 Characteristics of the patients, described with the 13 features.

The age at diagnosis and initiation of treatment is consistent with other studies of bipolar disorder using this data source (for example (Launders et al., 2022b) and EHR studies of bipolar disorder prescribing from other countries (Ng et al., 2021)).

Feature	Patients exposed to lithium	Patients exposed to olanzapine	
Age at diagnosis (ICD-10 criteria), median (IQR)	40.82 (24.13)	39.08 (22.36)	
Age at medication initiation, median (IQR)	46.51 (23.20)	42.47 (23.04)	
Years between diagnosis and exposure, median (IQR)	7.37 (15.60)	5.45 (13.35)	
Female, n (%)	11,526 (60.33 %)	6,858 (55.25 %)	
First presentation mania, n (%)	4,705 (24.63 %)	3,498 (28.18 %)	
First presentation depression, n (%)	11,233 (58.79 %)	7,453 (60.05 %)	
Depression dominant, n (%)	9,662 (50.57 %)	6,457 (52.02 %)	
Psychotic experiences, n (%)	4,939 (25.85 %)	3,806 (30.66 %)	
Self-harm history, n (%)	2,362 (12.36 %)	1,822 (14.68 %)	
Family history of bipolar disorder, n (%)	402 (2.10 %)	136 (1.10 %)	
Family history of depression, n (%)	375 (1.96 %)	245 (1.97 %)	
Family history of psychosis, n (%)	111 (0.58 %)	141 (1.14 %)	
Overweight or obese, n (%)	6,540 (34.23 %)	5,038 (40.59 %)	
Total, n	19,106	12,412	

Predictive modelling

We aim at differentially predicting response, here defined as successful monotherapy maintenance treatment, to lithium vs. olanzapine. For optimising prediction capacity, we considered three different ML algorithms; logistic regression with Elastic Net regularisation, Random Forest and Naïve Bayes. Elastic Net is a combined LASSO and Ridge penalised regression that uses both the l1 and the l2 norm in a proportion that can be relativised to the number of features in the resulting non-learning model. For large feature sets, we set the l1 norm at 80% weight, thus with the l2 norm at 20%; for small feature sets, the proportion was set to the inverse (Takada & Fujisawa, 2020). The β-coefficient associated with our predictors under study was then used to guide the search for variables that could play an important part in mechanisms for identifying non-responders. We randomly split data for both the cross-validation (five-fold stratified split) and for the separation into training and testing sets (stratified shuffle split). We focused on balanced accuracy as the most important metric for our models. This has the advantage of selecting a model based on the maximum true positive and true negative rate, appropriate for clinical applications (Kaivanto, 2008). Because no imputation was carried out, there is no risk of leaks from training to testing phase. Next, we used dimensionality reduction techniques to visualise and understand model output. By measuring our value space in two or three dimensions, we avoided problems of dimensionality that render simple measures in high-dimensional spaces uninformative (Aggarwal, Hinneburg & Keim, 2001). We carried out analyses to determine feature importance and augmented these with SHAP (Fig. S5).

ML experiments included variables from 31,518 unique patients. These patients were newly exposed to either lithium (n = 19,106) or Olanzapine (n = 12,412) and the distribution of our outcomes of interest was positive response (n = 11,848, 37.59%), negative response (n = 14,785, 46.91%) or equivocal response (n = 4,885, 15.50%). To characterise the sub-cohort of lithium responders (n = 8,017), we first used unsupervised ML to find low-dimensional separations between lithium responders and a comparison group of all lithium non-responders and all olanzapine-treated patients (n = 23,501). To preserve a high-dimensional separation when mapping to just two or three dimensions requires ocular inspection of a great number of candidates. Because of the stochastic nature of the procedure, we used it as indicative only of interesting separations of the latent space (Fig. 1). These indicative separations provide rudimentary visualisations of the unique features of lithium responders.

Figure 1 Potentially meaningful separations in the latent space, creating a reduced feature space.

The Uniform Manifold Approximation and Projection (UMAP) algorithm ran for 500 epochs, a fixed parameter for small graphs, and we wrote a script that generated images for 50, 70, 90, 110 and 130 neighbours, each with 20 different settings in 2D and another 20 in 3D. Out of the 200 different graphs generated, a dozen showed signs of meaningful separation between maintenance responders to lithium and other patients. For the above four examples, all in 3D but rendered here in 2D, were selected from this dozen images. All four are based on our set of 13 features with the largest support in the research literature.

To adhere to the openness requirement for replicability, our Python notebook of experiments is attached in the supplement, and we followed the STROBE reporting guidelines (Table S4). A data frame for the scikit-learn v1.0.1 library was set up in Python 3.9.10.

Results

To predict treatment response, we generated machine learning models using three methods: logistic regression with Elastic Net regularisation, Random Forest and naïve Bayes. Before running any experiments, the 5% of all patients with the most recent exposure were used to emulate a situation where the ML algorithm was dealing with new and completely unseen patients. To both cross-validate and use a holdout may seem straining, but we aim to keep adding data as it becomes available and having a model that scales to such unseen data is important.

Training and predicting on lithium patients only

Out of the three methods, Random Forest yielded the best prediction performance when trained on lithium patients only. Using 34 features, it scored a balanced accuracy of 0.616 and for the 13-feature set a very similar 0.614 (Table 2). We used balanced accuracy, the arithmetic mean of sensitivity and specificity, rather than F1-score, their harmonic mean, to account for our imbalanced classes. We then identified relative feature importance of the 34-feature set (Fig. 2A). The feature importance for models trained with Random Forest is based on the mean decrease in impurity. The top three features in importance were (1) age at BPD diagnosis, (2) age at time of first drug treatment, and (3) the time between diagnosis and treatment. Following these were features including mental well-being issues (depression, dominant mood, self harm), substance use (tobacco, alcohol) and metabolic problems (diabetes, overweight, high levels of low-density lipoprotein, hypothyroidism). Many features in this analysis proved to have very little contribution to the predictive power of the model.

Table 2 Quantitative experiment results when predicting any and every treatment response with different machine learning methods.

The balanced accuracy measures the arithmetic mean of sensitivity (true positive rate) and specificity (true negative rate), and since a random classifier scores 0.5, anything above that constitutes signal.

Number of features	Patient exposure	Subset of features	Balanced accuracy	
			Logistic regression	Random forest	Naïve Bayes	
34 features	Any	All features	0.589	0.513	0.600	
		Numerical only	0.586	0.585	0.589	
		Categorical only	0.531	0.500	0.538	
		Binary only	0.527	0.500	0.533	
	Lithium only	All features	0.604	0.616	0.596	
	Olanzapine only		0.577	0.500	0.595	
13 features	Any		0.592	0.556	0.591	
		Numerical only	0.586	0.583	0.589	
		Categorical only	0.500	0.500	0.503	
		Binary only	0.500	0.500	0.500	
	Lithium only	All features	0.597	0.614	0.591	
	Olanzapine only		0.557	0.500	0.581	

Figure 2 Feature importance in the best models predicting response, for lithium patients (left) and for all patients (right).

Training and predicting on olanzapine patients only

The Random Forest model for olanzapine patients failed entirely at predicting response (balanced accuracy 0.500). This indicates a structured difference worthy of attention, since Random Forest produces a cross-validated response difference, for both the 34- and 13-feature sets. This cohort also showed a weaker signal for logistic regression, than the cohort of lithium patients. For the more regularised naïve Bayes approach, things improved and for the 34-feature set, the olanzapine patients near-matched the lithium patients (see Table 2).

Training and predicting lithium maintenance response on all patients

For predicting who will stay on lithium for at least 2 years using a mix of lithium and olanzapine patients in model training, the best treatment response prediction was achieved by a model trained on all 34 features and using naïve Bayes, at 0.600 (again, see Table 2). From this model, we also identified the feature importance (Fig. 2B). For models trained with naïve Bayes, we used the permutation feature importance technique to inspect the model: this technique measures the decrease in a model score (here, balanced accuracy) when each single feature value is randomly shuffled. A high positive value for a feature shows that the performance is better when using real values for this feature instead of random values. A value close to zero (positive or negative) shows that there is little change in performance when using real values. We observe the same three most important features as for lithium response, although in a different order; other features being mainly mental well-being issues (stress, self-harm, psychosis, sleep problems).

Table 3 shows the performance of models predicting lithium maintenance responders after having used all patients for training. The model trained with naïve Bayes on 34 features displayed a very weak signal, with a balanced accuracy of 0.543. The feature importance for this model is shown in Fig. 3. To further attempt to train a model that could predict lithium responders, we employed a number of different methods based on the same cross-validation and stratification, including hyperparameter-optimised Random Forest ( 0.509) and LightGBM ( 0.528). In addition, we tested a very basic linear support vector classifier with UMAP transformation ( 0.510). With these results mostly in the negative, we conclude that more data as well as multi-site studies are needed to establish an ML-based classifier for lithium maintenance response, as part of a clinical decision support system.

Table 3 Quantitative experiment results when predicting only lithium maintenance response with different machine learning methods.

Number of features	Subset of features	Balanced accuracy	
		Logistic regression	Random forest	Naïve Bayes	
34 features	All features	0.510	0.500	0.543	
	Numerical only	0.501	0.500	0.517	
	Categorical only	0.500	0.500	0.500	
	Binary only	0.500	0.500	0.500	
13 features	All features	0.506	0.500	0.525	
	Numerical only	0.501	0.500	0.517	
	Categorical only	0.500	0.500	0.500	
	Binary only	0.500	0.500	0.500	

Figure 3 Feature importance in the best model for predicting lithium responders (balanced accuracy 0.600), trained with naïve Bayes on all 34 features.

Future research

With these results mostly in the negative, we conclude that more data as well as multi-site studies are needed to establish an ML-based classifier for lithium maintenance response, as part of a clinical decision support system. With our data sets from the UK, we are looking to compare to other countries and cultures by running our model on multi-site data. In an aggregated model with ensembling (Nunes et al., 2020a), we can then investigate the limits of the EHR data-driven approach. Besides getting closer to limits for what constitutes thresholds for useful and clinically actionable insights, such a generalisation would also test the scalability of our approach. If it scales as we anticipate, we have a transparent as well as relatively simple approach from a computational standpoint: we are still running shallow and classical epidemiological software.

Discussion

Clinical usefulness

Electronic Health Record data can inform clinicians if interpreted correctly and we have used it to provide new inferred information that could become part of future clinical decision support. Our analyses showed promise for the further characterisation of lithium responders using EHR data; the balanced accuracy of our best model was 0.61. However, our balanced accuracy was lower than in recent studies attempting to predict lithium response, notably not matching the so far largest prediction study (Nunes et al., 2020a) which had a balanced accuracy of 0.715 in their aggregated model. What is notable is that their aggregation resulted from a seven-site model with a much more ambitious formulation, whereas we used only EHR data. As a result of our, in this sense, much more transparent approach, our relatively modest balanced accuracy is good enough to try a large and multi-site sample, to which we have recently gained access. Moreover, in all models we were unable to predict who would respond well to olanzapine, indicating a structural difference in the data which is also worthy of further investigation.

We included a large cohort of individuals with BPD from a nationally representative population. We employed a range of ML models and carefully tested them in many experiments. We have reported in a transparent way the weak, but present, signal in the data and how it could be further processed using data-driven reasoning. The combination of supervised and unsupervised ML has been fruitful for a number of healthcare applications, not least because of its approach to controlling bias (Launders et al., 2022a). EHR records do not include the detailed information that would facilitate a nuanced understanding of an individual’s treatment response in bipolar disorder such as optimisation of their mental state or level of functioning. Instead, we defined responders as patients treated for two or more years with no need for additional psychotropic medication. This outcome is therefore a proxy for a combination of effectiveness and tolerability of the drug because it indicates an individual has not needed alternative or additional treatments. This is the approach taken in pragmatic trials of maintenance treatments for BPD (Geddes et al., 2009). The proportion of patients that met this criterion in our cohort is similar to that of previous studies (Garnham et al., 2007; Alda, 2017). Other clinical features that may be of interest are not routinely recorded in the EHR, such as the phasic nature of the patients affective symptoms (Hui et al., 2019), or may only be recorded when there is an indication for doing so, such as inflammatory blood markers when infection is suspected.

ML approaches may constitute a black box where the primary aim is a good prediction rather than a complete understanding of the data and less concern with epidemiological considerations regarding causation, bias, confounding, multicollinearity, censoring and truncation. The causal relation between treatment selection and maintenance treatment response for an individual is an important, but separate research question. In spite of our use of propensity scores for cohort inclusion, stratification, and optimization of our experiments, it is likely that some bias in treatment allocation remains; for example, it has previously been shown that clinicians are more likely to prescribe olanzapine to individuals with lower baseline weight (Bazo-Alvarez et al., 2020). Unfortunately, because of the coding system used in the UK (ICD-10 and Read codes) we were not able to stratify our results by bipolar disorder subtype, or consider this as a variable in the models. As we noted under Results above, scalability is also of the greatest importance for adoption at the clinic.

Validity of the findings

The primary reason for why the covariates vary, as compared to published research, is that we use only EHR data. A secondary reason is probably that we use UK data only. We will in the future analyse multi-site EHR data, which we have recently received access to. One may further ask why the features with highest relative importance are likely predictors. The relative importance found in our data-driven approach denotes the amount of variance explained. We thus adhere to classical machine learning explorative studies into the positive predictive value of features.

Agnostic data-driven modelling

The agnostic part of our modelling has the potential to unravel EHR data point associations that could point to, albeit not themselves constitute, causal explanations. We addressed our prediction problem both in an informed (domain expert) way and in an agnostic (layperson) way. We therefore had the choice of (a) letting domain experts drop predictive features that are, according to them, likely artefacts, (b) letting domain experts consider predictive features as novel variables helpful to understand clinical association, or (c) both of the former. We went with (c), as our project has been set up as an interdisciplinary study, to maximise the chance of providing correct clinician decision support by involving multiple perspectives (data-driven as well as hypothesis-driven, conservative as well as progressive with respect to the field).

Predictive models using data sources such as EHRs can provide real world outcomes information for large numbers of patients. Given the increasing number of covariates available in these data sources, flexible and data-driven machine learning procedures may usefully complement hypothesis-driven traditional regression methods, because of the ability to discover hidden predictors and interactions, nonlinear, and higher-order effects, as well as to approximate intricate functions poorly represented by individual covariate terms or interaction terms (Chekroud et al., 2021). Potentially pairing predictions from EHRs with those from genetic and neuroimaging data may become clinically feasible in the near future (Ho et al., 2020; Claude et al., 2020). The clever combination of completely data-driven results, gained from unsupervised machine learning with decision trees, with a less explorative and more informed perspective, gained from supervised machine learning fusion techniques (Carbonell, Boman & Laukka, 2021) is what enables us to do efficient agnostic modelling.

Conclusions

We used a variety of ML models and techniques to assess treatment response to lithium and olanzapine in BPD using data from clinical electronic records. We found differences between patients receiving maintenance treatment with lithium and the other patients in our cohort, with respect to prediction of a good response over 2 years. However, we were unable to predict who would respond well to olanzapine using these techniques. Even if we failed to predict successful monotherapy olanzapine treatment, and so to definitively separate lithium vs. olanzapine responders, the characterization of the two groups can be used as a classifier. This classification by proxy can, in turn, be useful for establishing maintenance therapy. That said, our results needs to be corroborated and hopefully more pronounced, in an even larger sample. Overall, our findings support the further exploration of supervised and unsupervised ML methods using EHRs for drug treatment selection as part of clinical decision support.

Supplemental Information

Supplemental Information 1 Pearson (left) pairwise linear relationships between continuous variables and Spearman (right) monotonic relationship based on rank.

Supplemental Information 2 Chi-squared matrix for all pairs of categorical variables. Variables are ordered by importance, as informed by published research literature.

Supplemental Information 3 Distributions of the two variables most important to the primary outcome, for the 31,518 patients with more than two years of exposure.

Supplemental Information 4 Plots of values of the year variable for age first exposure (age at exposure start), age first diagnosis and symptom to exposure (the number of years from first symptom to first exposure to Olanzapine or lithium).

Supplemental Information 5 Age at diagnosis, established using ICD-10 criteria.

Supplemental Information 6 SHAP (Shapley additive explanations) summary statistics plot indicating the Shapley values for the 13-feature set with features in descending order of importance, when predicting if a patient is responding to lithium.

Supplemental Information 7 The data dictionary for the original full set of 119 variables (including the outcomes) elicited from the EHRs.

Supplemental Information 8 Binary variable correlations to the target variable (response), measured with the Simple Matching Coefficient.

Supplemental Information 9 Conditional entropy for a curated set of variables to potentially turn into machine learning features.

Supplemental Information 10 STROBE protocol checklist.

Supplemental Information 11 Supplemental Information.

Additional Information and Declarations

Competing Interests

Author Contributions

Ethics

Data Availability

Joseph F. Hayes has received consultancy fees from juli Health and Wellcome Trust. No other authors declare competing interests.

Joseph F. Hayes conceived and designed the experiments, performed the experiments, analyzed the data, prepared figures and/or tables, authored or reviewed drafts of the article, and approved the final draft.

Fehmi Ben Abdesslem performed the experiments, analyzed the data, prepared figures and/or tables, authored or reviewed drafts of the article, and approved the final draft.

Sandra Eloranta analyzed the data, authored or reviewed drafts of the article, and approved the final draft.

David P. J. Osborn conceived and designed the experiments, authored or reviewed drafts of the article, and approved the final draft.

Magnus Boman performed the experiments, analyzed the data, prepared figures and/or tables, authored or reviewed drafts of the article, and approved the final draft.

The following information was supplied relating to ethical approvals (i.e., approving body and any reference numbers):

We completed a cohort study using primary care EHR data collected between January 1, 2000, and December 31, 2018, by the Clinical Practice Research Datalink (CPRD) system. The study was approved by the Independent Scientific Advisory Committee of CPRD and the data was extracted in April 2019.

SAC of CPRD protocol no. 18_316. CPRD HRA UK: East Midlands Derby Research Ethics Committee 21/EM/0265.

CPRD obtains annual research ethics approval from the UK Health Research Authority Research Ethics Committee (East Midlands—Derby Research Ethics Committee reference number 05/MRE04/87) to receive and supply patient data for public health research. Therefore, no additional ethics approval was required for this study.

The following information was supplied regarding data availability:

The prediction model code in Python (Jupyter Notebook) is available at GitHub:

- github.com/fehmi8/lithium

- Fehmi. (2024). fehmi8/lithium: Predicting maintenance lithium response for bipolar disorder from electronic health records—a retrospective study (peerj). Zenodo. https://doi.org/10.5281/zenodo.12806418.

The data that support the findings of this study are available from CPRD but restrictions apply to the availability of these data, which were used under license for the current study, and so are not publicly available. Electronic health records are, by definition, considered to be sensitive data in the UK by the Data Protection Act and cannot be shared via public deposition because of information governance restriction in place to protect patient confidentiality. Access to data is available only once approval has been obtained through the individual constituent entities controlling access to the data.

The primary care data can be requested via application to the CPRD. Details of how to access the data can be found here: https://cprd.com/data-access. Data access is subject to protocol approval via CPRD’s Research Data Governance (RDG) Process.

The study was approved by the Independent Scientific Advisory Committee of Clinical Practice Research Datalink (CPRD) system, protocol no. 18_316, and the data was extracted in April 2019.

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
