# Peer review of "Predicting maintenance lithium response for bipolar disorder from electronic health records—a retrospective study"

_PeerJ, doi:10.7717/peerj.17841_

## Round 0.1 · original submission · Major Revisions

I have now received the reviewers' comments on your manuscript. They have suggested some revisions to your manuscript. Therefore, I invite you to respond to the reviewers' comments and revise your manuscript.


Reviewer 1 ·

Basic reporting

no comment

Experimental design

1. INTRODUCTION part: The authors need to present more background on olanzapine monotherapy in BPD and explain why olanzapine monotherapy can serve as a control group in this study.

2. METHODS: Why authors defined treatment response as continuous prescription of the study drug (lithium or olanzapine) for at least two years without stopping treatment? The treatment response used by other ML studies mentioned in the article are not this(Kim et al., 2019; Nunes et al., 2020).

3.line 173: "...(Figure ??)..." please update.

4.line183: "(b) lithium responders: whether a patient is a lithium responder or not." what is the definition of lithium responder?

Validity of the findings

he authors need to add additional discussion on why the covariates found in this study differ from previous studies. And why the most relative importance features are possible the predictors and whether there is a clinical or mechanistic association

·

Basic reporting

Hayes et al. research response to olanzapine or lithium in UK electronic health records. The language of the manuscript is fine; the introduction covers most of the relevant literature. On p. 7, a figure is not labelled (“??”). The question is of high interest to the field of mental disorders in general, and to bipolar disorder in particular. However, I feel that the manuscript is lacking structure. Many analyses were carried out, but the results part is only very short (see also below).

The research question and analyses to answer these questions appear appropriate, but I find the paper very hard top read. Many analyses were carried out, but a true structure that guides the reader is absent. There should be many more subheadings, dividing the analyses into meaningful parts. For example, the question stated in the introduction is “to determine if ML models could differentially predict response to lithium monotherapy vs. olanzapine”. However, later in the manuscript the authors state that the question is to predict “(a) response: whether a patient will stay on a treatment for more than 2 years, and (b) lithium responders”.

Experimental design

To me, it is not entirely clear how the unsupervised part (Agnostic feature selection and clustering) relates to the supervised/classification part of the analysis. This may all be well-justified, but the authors should extend the results part, to make it easier for the reader to follow. Also, both the results and the discussion parts should be longer, i.e. give more information, given that many analyses were carried out.

Validity of the findings

I will be able to assess this once the manuscript has been structured better.

Additional comments

The authors mention that “olanzapine is included in several guidelines as an effective maintenance treatment for BPD”. Could the authors give a literature reference?

Reviewer 3 ·

Basic reporting

Raw data cannot be shared.

Clear English language used throughout. However, some parts can be clarified.

In the Abstract:

- What is case and what is control? Can you provide additional information?
- Is there a difference between maintenance responders and responders (lithium maintenance responders and olanzapine responders)? If not, please use the same terminology. If there is a difference, please describe this.
- It is unclear to me to what ‘problem’ the authors refer exactly. Could this be explained?

In the Introduction:

- It would be helpful to provide a clear definition of lithium response in relation to the literature that is described. Studies use different definitions, and that should be described here, as it also helps the interpretation of the author's own study results.

In the Methods:

- Line 173. It says Figure??

In the Discussion:

- Line 292-293. "Even if we failed to predict who would respond to olanzapine and to definitively separate lithium vs olanzapine responders, that problem is so difficult...". I don't understand what is meant by "that problem". Please rephrase.

- Line 252. It would help the reader if the authors would provide information whether a balanced accuracy of 0.61 is acceptable for ML models. Or what would be a balanced accuracy that is acceptable, as other studies showed models with a higher balanced accuracy.

Last, I am wondering whether maintenance treatment response really is the correct terminology, as it suggests the authors have information about the treatment response (e.g., no mood episodes, no hospitalization, etc). Wouldn’t it be better to say prediction of lithium or olanzapine maintenance. Because that is what the authors actually measure.

Experimental design

There is one research question, "Our aim was to determine if ML models could differentially predict ¨ response to lithium monotherapy vs. olanzapine monotherapy in an incident cohort of people with BPD". This is a clear research question. However, in the text it seems that 2 research questions are answered. For instance in the conclusions, where the prediction of lithium and olanzapine maintenance AND the separation of lithium from olanzapine responders is described. Should the separation of lithium from olanzapine responders be a second research question?

Also, the Methods section should include more clinical information.

- Can the authors provide more information on the assessment of bipolar disorder in this cohort? How were patients diagnosed. Was the diagnosis based on the DSM or ICD?
- What is meant by stress (one of the features that is added to the model)? How was this measured?
- Table 1 shows the median age at diagnosis and age at initiation of medication. This is much higher than often reported in the literature. Could the authors reflect on this?
- In the methods (line 97) it is described that Olanzapine is used as a comparison group. For what? For Lithium maintenance? Olanzapine maintenance is also an outcome of the research question.
- There is currently no information on subtype of bipolar disorder. Is this not available in the data? It is possible that prediction of lithium or olanzapine maintenance is different for individuals with BD-I, BD-II and BD-NOS. It is important to include this information if it is available. And if not, to reflect on this in the discussion.

Validity of the findings

- EHR data provide much information on large sample sizes and are very informative to investigate the research question of the authors. It would be very interesting to see if results would differ for BD-I and BD-II patients. Also, would there be differences for patients with a younger age at diagnosis compared to patients with an older age at diagnosis.

Additional comments

This is an interesting study that contributes to the current literature on lithium and olanzapine use. However, several points need to be clarified.

---

## Round 0.2 · Major Revisions

Thank you for the update. However, there are still concerns that prevent me from accepting the revised paper. Please pay attention to the reviewers' comments and respond to them carefully.

Reviewer 1 ·

Basic reporting

no comment

Experimental design

no comment

Validity of the findings

no comment

·

Basic reporting

I am sorry but I do not think that the authors have improved the structure of their manuscript significantly. As far as I can see, the authors conducted two main analyses. Response to lithium -> yes/no AND response to olanzapine -> yes/no. In the results part, I would expect to have one subheading "Prediction of Response to Lithium" and another one "Prediction of Response to Olanzapine". It should also be mentioned that (as far as I understand) a third analysis was carried out that, for any patient, tried to classify her/him as an olanzapine or lithium responder. Also, in the abstract, the authors mention case and control, without defining clearly what they mean by this. While I really do think that the authors have carried out valuable and highly skilled analyses, I also think that the analyses must be explained better. For me, this is crucial.

Experimental design

The authors state that olanzapine responders were not in their holdout sample (5% of patients with most recent exposure). I wonder why the authors did not change the size of their holdout sample to 10 or 20%? To me, it seems that there are enough data, and the results would be very important.

Validity of the findings

No comment

Reviewer 3 ·

Basic reporting

The manuscript has improved. The authors have included information that improve the readability of the manuscript. However, in my opinion, some parts could be better moved to another part of the manuscript. Also, some parts still need additional information.

- In the new part in the introduction: “During the period covered by this study, the National Institute for Health and Care Excellence (UK) recommended lithium, sodium valproate or olanzapine as first-line maintenance monotherapy FOR BPD”. Please add for BPD to the sentence.
- Also, the last sentence: “So we believe the two groups are comparable”, does not fit well in this part of the introduction section. Could this be rephrased; e.g., these findings suggest that patients on monotherapy for lithium and olanzapine are comparable…”. And what do the authors mean exactly by comparability? Comparable in size? In clinical characteristics? In efficacy of the treatment?
- In addition, I think the new part of the discussion is important and a good addition:
"We defined successful monotherapy maintenance treatment as treatment for at least two years, without switching medication or the addition of new psychotropic medications. Our definition therefore suggests the patient found the medication effective and tolerable. We believe this is a close approximation of treatment response as defined by the Alda Scale [New REF https://pubmed.ncbi.nlm.nih.gov/12416605/] and used in previous studies predicting lithium response. We see a similar proportion of “responders” in our cohort to earlier studies [Kim et al. 2019, Nunes et al 2020]. The Alda scale is frequently dichotomized, with scores >6 representing treatment response; this score is challenging to achieve if the duration of treatment is less than two years and if additional psychotropic medication is required. Therefore, our patients staying on treatment for over two years without additional medication will be similar to those identified as responders by the “gold standard” clinician rated scale [New REF https://www.ncbi.nlm.nih.gov/pmc/articles/PMC6984707]. Given the complexity of the biology of successful lithium maintenance treatment [Papiol et al. 2022], we believe this makes our prediction results commensurable to the previous literature".

- However, for the flow of the paper, I wonder whether it’s not better to integrate this with the text in the method section, and then reflect on this in the discussion section.
- The main results should be described in the first paragraph of the Discussion section. With the current additions in the discussion, this is not the case. The authors should consider moving the paragraph Clinical Usefulness to the beginning of the discussion and clearly describe what the main finding of the study is.

Experimental design

No comment

Validity of the findings

This sentence has been improved in readability: "Even if we failed to predict successful monotherapy olanzapine treatment, and so to definitively separate lithium vs olanzapine responders, the characterization of the two groups can be used as a classifier. This classification by proxy can, in turn, be useful for establishing maintenance therapy". However, based on the results, I wonder whether the authors can really say that the characterization of the two groups can be used as a classifier. This conclusion seems too strong. There is some evidence pointing towards prediction of lithium monotherapy, but not olanzapine therapy.

---

## Round 0.3 · Major Revisions

Thank you for submitting the revised version of the manuscript. While most of the reviewers' concerns have been properly addressed, I have a major concern that needs attention. Can the authors provide more details regarding the hyperparameter tuning process? It is crucial to ensure models are neither underfitted nor overfitted, and this process should be handled with caution. Bayesian search is generally preferred, and the details of the final hyperparameter settings should be reported. Additionally, the authors mentioned that they attached their Python code in the supplementary materials; however, I was unable to locate it.

·

Basic reporting

The authors have now explained the study design better. Thank you!

Experimental design

no comment

Validity of the findings

no comment

Reviewer 3 ·

Basic reporting

- More explanation on results is now provided, which is helpful for the interpretation.
- The Future Research paragraph fits better in the Discussion section than in the Results section.

Experimental design

Clear

Validity of the findings

Clear

---

## Round 0.4 · Major Revisions

Thank you for uploading your Python code. Upon review, it appears that the hyperparameter configuration for your dataset, which includes over 10,000 subjects and more than 100 features, might be suboptimal. For example, the upper bound of n_estimators could be increased to 500 or even 1,000 to potentially improve performance. Could you please confirm this issue? If necessary, I recommend consulting with a data scientist for further optimization.

---

## Round 0.5 · accepted · Accept

Thank you for your effort in addressing all the concerns.